# Prognostic Impact of the HFA-PEFF Score in Patients with Acute Myocardial Infarction and an Intermediate to High HFA-PEFF Score

**DOI:** 10.3390/jcm11154589

**Published:** 2022-08-05

**Authors:** Kwan Yong Lee, Byung-Hee Hwang, Chan Jun Kim, Young Kyoung Sa, Young Choi, Jin-Jin Kim, Eun-Ho Choo, Sungmin Lim, Ik Jun Choi, Mahn-Won Park, Gyu Chul Oh, In-Ho Yang, Ki Dong Yoo, Wook Sung Chung, Kiyuk Chang

**Affiliations:** 1Cardiovascular Center and Cardiology Division, Seoul St. Mary’s Hospital, The Catholic University of Korea, Seoul 06591, Korea; 2Cardiovascular Center and Cardiology Division, Uijeongbu St. Mary’s Hospital, The Catholic University of Korea, Uijeonbu 11765, Korea; 3Cardiovascular Center and Cardiology Division, Yeouido St. Mary’s Hospital, The Catholic University of Korea, Seoul 07345, Korea; 4Cardiovascular Center and Cardiology Division, Incheon St. Mary’s Hospital, The Catholic University of Korea, Incheon 21431, Korea; 5Cardiovascular Center and Cardiology Division, Daejeon St. Mary’s Hospital, The Catholic University of Korea, Daejeon 34943, Korea; 6Department of Cardiovascular Medicine, Kyung Hee University Hospital, Seoul 05278, Korea; 7Cardiovascular Center and Cardiology Division, St. Vincent’s Hospital, The Catholic University of Korea, Suwon 16247, Korea

**Keywords:** heart failure with preserved ejection fraction, myocardial infarction, percutaneous coronary intervention, heart failure

## Abstract

This study aimed to investigate the efficacy of the HFA-PEFF score in predicting the long-term risks in patients with acute myocardial infarction (AMI) and an HFA-PEFF score ≥ 2. The subjects were divided according to their HFA-PEFF score into intermediate (2–3 points) and high (4–6 points) score groups. The primary outcome was all-cause mortality. Of 1018 patients with AMI and an HFA-PEFF score of ≥2, 712 (69.9%) and 306 (30.1%) were classified into the intermediate and high score groups, respectively. Over a median follow-up of 4.8 (3.2, 6.5) years, 114 (16.0%) and 87 (28.4%) patients died in each group. Multivariate Cox regression identified a high HFA-PEFF score as an independent predictor of all-cause mortality [hazard ratio (HR): 1.53, 95% CI: 1.15–2.04, *p* = 0.004]. The predictive accuracies for the discrimination and reclassification were significantly improved (C-index 0.750 [95% CI 0.712–0.789]; *p* = 0.049 and NRI 0.330 [95% CI 0.180–0.479]; *p* < 0.001) upon the addition of a high HFA-PEFF score to clinical risk factors. The model was better at predicting combined events of all-cause mortality and heart failure readmission (C-index 0.754 [95% CI 0.716–0.791]; *p* = 0.033, NRI 0.372 [95% CI 0.227–0.518]; *p* < 0.001). In the AMI cohort, the HFA-PEFF score can effectively predict the prognosis of patients with an HFA-PEFF score of ≥2.

## 1. Introduction

Heart failure is a clinical syndrome in which the sufficient oxygen needed for peripheral organs is not delivered due to structural or functional abnormalities of the ventricle rather than a specific disease. There are 64 million heart failure patients worldwide, and 40% to 50% are reported to have heart failure with a preserved ejection fraction (HFpEF) [1,2]. The prevalence of HFpEF is increasing as life expectancy increases, and the population is aging. The increase in the prevalence of comorbid diseases such as coronary artery disease, atrial fibrillation, obesity, metabolic syndrome, and diabetes also contributes to this [3]. The mortality rate of HFpEF is lower than that of heart failure with a reduced ejection fraction (HFrEF) but higher than that of the general population of the same age [4]. Providing effective management is a major unmet clinical need for HFpEF patients, who depend on a clear diagnosis and identification of predictors associated with the condition.

The 2021 European Society of Cardiology (ESC) guidelines have designated HFpEF as a left ventricular ejection fraction (LVEF) of ≥50% when there is objective evidence of cardiac structural and/or functional abnormalities with symptoms of heart failure [5]. HFpEF should not be excluded by the cutoff value of the single general parameter because there are various risk factors and phenotypes due to comorbidities. The Heart Failure Association of the European Society of Cardiology proposed the HFA-PEFF diagnostic algorithm for HFpEF in a consensus report in 2019 [6]. 

The HFA-PEFF score involves screening for functional or morphological abnormalities through echocardiography and measuring the NT-proBNP level; two points are given if each major criterion is satisfied, and one point if the minor criterion is satisfied. If the total HFA-PEFF score is less than one point, the probability of a diagnosis of HFpEF is unlikely, and an investigation into alternative causes of the disease is necessary. When the score is higher than five points, an immediate diagnosis of HFpEF is recommended. With a score of 2–4 points, a diastolic stress test or invasive hemodynamic measurements are recommended for a definitive diagnosis of HFpEF. The HFA-PEFF score has been well-validated for the diagnosis of HFpEF. However, its predictive power remains unclear. In addition, the HFpEF of ischemic etiology has a worse prognosis than that of nonischemic etiology, and the role of the HFA-PEFF score in the AMI population is expected to be highly important but has not been reported to date. Therefore, we aimed to investigate the predictive value of the score for patients with an HFA-FEFF score of ≥2 in our cohort with acute myocardial infarction.

## 2. Materials and Methods

### 2.1. Study Protocols and Population Selection 

The Convergent Registry of Catholic and Chonnam University for Acute MI (COREA-AMI) registry was designed to evaluate real-world, long-term clinical outcomes in all consecutive patients with AMI at nine major cardiac centers in Korea. All of the hospitals perform a large number of percutaneous coronary interventions (PCIs) in AMI patients and are located throughout the country. The COREA-AMI I registry included AMI patients undergoing PCI from January 2004 to December 2009, and the COREA-AMI II registry extended the follow-up period of COREA-AMI I patients and enrolled additional AMI patients from January 2010 to August 2014. The clinical, angiographic, and follow-up data of all AMI patients were consecutively registered in the electronic, web-based case report system. The COREA-AMI study was conducted in accordance with the Declaration of Helsinki. This observational study was approved by the Catholic Medical Center Central Institutional Review Board (IRB) and each participating hospital IRB (IRB No.XC15RSMI0089K). In addition, the study was performed in accordance with the Strengthening the Reporting of Observational Studies in Epidemiology guidelines [7]. The COREA-AMI registry is registered on ClinicalTrials.gov (study ID: NCT02806102).

In total, 10,719 patients with AMI who underwent PCI with drug-eluting stents (DESs) were enrolled in the registry, and patients who did not undergo NT-proBNP testing at admission (N = 4012), those missing the echo data during hospitalization (N = 393), those who underwent echocardiography before revascularization (N = 1881), those with an LVEF < 50% (N = 2836), and those with a low HFA-PEFF score of ≤1 (N = 579) were excluded from the analysis. Thus, 1018 patients were selected for this analysis. A study flowchart is depicted in Figure 1. The objective of the present study was to evaluate the efficacy of the HFA-PEFF score in predicting the long-term risks following revascularization in patients with AMI who were suspected of HFpEF. The definitions adopted for each HFA-PEFF criterion in this study are shown in Appendix A. The patients were classified into intermediate HFA-PEFF score (2–3 points) or high HFA-PEFF score (4–6 points) groups according to the criteria from the 2019 consensus report [6].

### 2.2. PCI Procedure and Medical Treatment 

All of the patients underwent PCI within 48 h after admission. Coronary angiography and primary PCI were performed according to the current standard guidelines. Significant coronary artery disease was defined by angiographic stenosis ≥70% in the epicardial coronary arteries and ≥50% in the left main coronary artery. A loading dose of the antiplatelet agent (aspirin, 300 mg; clopidogrel, 300 mg or 600 mg; cilostazol, 200 mg; ticagrelor, 180 mg; or prasugrel, 60 mg) was prescribed for all patients before or during PCI. Patients with DESs were prescribed P2Y12 inhibitors (clopidogrel, 75 mg once daily; ticagrelor, 90 mg twice daily; or prasugrel, 10 mg once daily) and/or aspirin, 100 mg daily. The duration of dual antiplatelet agent administration was determined by a physician in accordance with the final diagnosis at baseline and the revascularization procedure complexity. Optimal pharmacological therapy, including statins, beta-blockers, angiotensin-converting enzyme (ACE) inhibitors, or angiotensin II receptor blockers (ARBs), was recommended according to the guidelines. The doses were titrated, and the medications were changed during the follow-up if needed, depending on each patient’s condition. Predilation, direct stenting, postadjunct balloon inflation, and glycoprotein IIb/IIIa receptor blocker administration were performed at the discretion of individual physicians.

### 2.3. Study Endpoints and Follow-Up 

The primary endpoint of this analysis was all-cause mortality at 5 years after index PCI for AMI. The secondary endpoints were cardiovascular death, recurrent MI, ischemic stroke, any revascularization, target vessel revascularization (TVR), target lesion revascularization (TLR), and overt bleeding (Bleeding Academic Research Consortium [BARC] type 3 or 5) [8]. Cardiovascular death was defined as death resulting from AMI, sudden cardiac death, heart failure, stroke, or other vascular causes. Ischemic stroke was defined as an episode of neurologic dysfunction related to the brain, spinal cord, or retinal vascular injury because of infarction. Each patient was followed up at outpatient clinics or by telephone questionnaire at 1, 6, and 12 months and then annually thereafter. All of the data were collected in a web-based system after eliminating personal information. Patient follow-up data, including survival data and clinical event data, were collected through 31 March 2019 via hospital chart reviews and telephone interviews of patients conducted by trained reviewers who were blinded to the study results. Independent reviewers and interventional cardiologists assessed angiographic and procedural data, and independent research personnel collected baseline clinical, laboratory, and medication data. All adverse clinical events of interest were confirmed centrally by the committee of the Cardiovascular Center of Seoul St. Mary’s Hospital (Seoul, Korea). The validation of mortality was performed on the basis of disqualification from the National Health Insurance Service, which is the single government-managed insurance provider and covers almost all of the nation’s population. The final dataset was handled by independent statisticians at the clinical research coordinating center and sealed with a code by the clinical research associate.

### 2.4. Statistical Analysis 

The categorical variables were presented as numbers and relative frequencies (percentages) and were compared using the chi-squared test or Fisher’s exact test. The continuous variables were expressed as the mean ± standard deviation or median (Q1, Q3), depending on whether they were normally distributed or not, and were compared using the independent sample t-test or Mann–Whitney U test, as appropriate. A D’Agostino–Pearson test was conducted to evaluate whether continuous variables were normally distributed or not. The cumulative event rates of each group were calculated using a Kaplan–Meier estimator and compared using the log-rank statistic. All of the parameters showing a *p*-value of <0.05 in univariable analysis and known clinical risk factors for HFpEF were included in multivariable analysis. The adjusted variables for the multivariate Cox proportional hazard regression analysis were age ≥75, female sex, diabetes, hypertension, atrial fibrillation, and chronic kidney disease. To identify independent echocardiographic and clinical predictors of all-cause mortality, we used a multivariable Cox proportional hazard model. Two prediction models were constructed to assess the incremental prognostic value of high HFA-PEFF scores (≥4) and applied to all-cause mortality data: (1) Model A: conventional clinical risk factors; (2) model B: model A + high HFA-PEFF scores. Moreover, models A and B were applied to the composite outcomes of all-cause mortality and heart failure readmission. Conventional clinical risk factors were based on the published risk factors for symptomatic HFpEF (2019 EHJ HFA-PEFF). The discriminative ability of the models was assessed using Harrell’s C-index, which is analogous to the area under the receiver operator characteristic curve and was applied to all-cause mortality data. Reclassification performance was compared using the relative integrated discrimination improvement (IDI) and category-free net reclassification index (NRI). Each measure was analyzed using R version 4.1.2 (R Foundation for Statistical Computing, Vienna, Austria). Statistical significance was indicated by a two-tailed *p* < 0.05.

## 3. Results

### 3.1. Baseline Characteristics 

The baseline clinical, echocardiographic, and angiographic characteristics are listed in Table 1 and Table 2. The mean age of all included patients was 65.0 ± 12.3 years, and 33.0% were female. Overall, 31.6% had diabetes mellitus, 59.3% had hypertension, and 7.1% had a previous stroke. Regarding angiographical lesion and procedural profiles, only 3.6% of the patients presented with cardiogenic shock. Of the 1018 included patients, 712 patients were classified into the intermediate HFA-PEFF score group, and 306 were classified into the high HFA-PEFF score group. The patients in the high HFA-PEFF score group were more likely to be older and female, have diabetes mellitus, have hypertension, have chronic kidney disease, and use diuretics more often. Regarding the laboratory data, the patients in the high HFA-PEFF score group had a higher level of N-terminal pro B-type natriuretic peptide (NT-proBNP), creatinine, and high sensitivity troponin but lower levels of hemoglobin, total cholesterol, triglycerides, high-density lipoprotein cholesterol, and low-density lipoprotein cholesterol than those in the intermediate HFA-PEFF score group. Interestingly, the discrepancy in the level of natriuretic peptides was mainly observed in patients with sinus rhythm (*p* < 0.001) but not in those with atrial fibrillation (*p* = 0.449). 

Regarding the echocardiographic findings, the high HFA-PEFF score group had higher values of the left ventricular end-systolic diameter, left ventricular end-diastolic diameter, mitral Doppler early velocity/mitral annular early velocity (E/e’), and pulmonary artery systolic pressure but lower values of the LVEF (Table 2). 

For the biomarker domain of the HFA-PEFF score, the majority of the patients (76.0%) received two points, whereas 32.1% and 37.6% of the patients received two points for the functional and morphological domains, respectively (Table 3).

### 3.2. Clinical Outcomes

Over a median follow-up of 4.8 (3.2, 6.5) years, 114 (16.0%) and 87 (28.4%) patients died in the intermediate and high score groups, respectively. Kaplan–Meier analysis showed that the all-cause mortality rate was significantly higher in the high score group (28.4%) than in the intermediate score group (16.0%) (*p* < 0.001) (Figure 2, Table 4). The difference in the results was mainly in cardiovascular death (21.6% vs. 12.1%, *p* < 0.001). The readmission rate due to heart failure was also significantly higher in the high score group (6.5%) than in the intermediate score group (1.5%) (*p* < 0.001). A concordant result was shown in the sensitivity analysis, which adjusted for the influence of multivariate variables (Table 4). The multivariate Cox regression demonstrated that age ≥75 (HR: 4.33, 95% CI 3.23–5.8, *p* < 0.001), chronic kidney disease (HR: 3.97, 95% CI: 2.11–7.5, *p* < 0.001), atrial fibrillation (HR: 2.12, 95% CI: 1.04–4.33, *p* = 0.039), diabetes mellitus (HR: 1.81, 95% CI: 1.35–2.42, *p* < 0.001), hypertension (HR: 1.4, 95% CI: 1.01–1.92, *p* = 0.042) and a high HFA-PEFF score of ≥4 points (HR: 1.53, 95% CI: 1.15–2.04, *p* = 0.004) were significant predictors of all-cause mortality after adjustment (Table 3). Among the three components (biomarker, functional, and morphological score) that make up the HFA-PEFF score, a high biomarker score (≥2) showed the highest risk of all-cause mortality (HR: 1.98, 95% CI 1.3–3.03, *p* = 0.002) (Table 3).

Kaplan–Meier curves with cumulative hazards of all-cause mortality between the intermediate and high HFA-PEFF score groups.

Receiver operating characteristic analysis was performed to evaluate the ability of the HFA-PEFF score to predict mortality in patients with an HFA-PEFF score of ≥2 in the AMI cohort, with an area under the curve (AUC) of 0.59 (95% CI: 0.56–0.62, *p* < 0.001). The sensitivity and specificity of the HFA-PEFF score at a cutoff of ≥ 4 were 43.3% and 73.3%, respectively. The addition of a high HFA-PEFF score (≥4) to clinical risk factors (model B), such as old age, female sex, hypertension, diabetes mellitus, atrial fibrillation, and chronic kidney disease, significantly increased the discriminant ability to predict mortality compared with that of the risk factors alone (model A) (C-index: 0.750, 95% CI: 0.712–0.789, *p* = 0.049) (Figure 3A and Table 5). For predicting mortality, model B also showed a significantly higher reclassification ability than model A (NRI: 0.330, 95% CI: 0.180–0.479, *p* < 0.001). When predicting the composite of mortality and readmission due to heart failure, the additive model showed the highest C-index (0.754, 95% CI: 0.716–0.791), with significant improvement over the conventional clinical risk factors only model (*p* = 0.033). For predicting the composite events, the HFA-PEFF score addition model showed a significantly higher reclassification ability against the conventional risk factors model (NRI: 0.372, 95% CI: 0.227–0.518, *p* < 0.001; IDI: 0.007, 95% CI: 0–0.014, *p* = 0.047) (Figure 3B and Table 5).

## 4. Discussion

In the present study, we compared the five-year clinical outcomes of patients with AMI between the high HFA-PEFF score group versus the intermediate HFA-PEFF score group using data from a large multicenter observational study. The main findings were as follows. First, the optimal cutoff value of the HFA-PEFF score is ≥4 for mortality. Second, the high HFA-PEFF score group showed a significantly higher mortality risk than the intermediate HFA-PEFF score group (Figure 2). Third, the HFA-PEFF score can effectively predict the prognosis not only in the general population but also in the AMI cohort (Figure 3). 

### 4.1. Pathophysiology and Comorbidities of HFpEF

Heart failure with ischemic etiology is known to have a higher mortality rate than that with nonischemic etiology [9]. In particular, for HFpEF with previous myocardial infarction (MI), cardiovascular mortality and sudden cardiac death are reported to be significantly higher than that without MI [10]. Based on the mechanism of HFpEF, myocardial dysfunction and structural remodeling are driven by endothelial oxidative stress, and multiple diastolic abnormalities in cardiovascular function may contribute to HFpEF development [11,12]. However, in practice, it is very difficult to specify the mechanism of HFpEF. It is generally considered to be a clinical phenotype of the outcome of a combination of multiple risk factors and comorbidities [13]. In this study, the included subjects also had a variety of comorbidities, such as high blood pressure (59.3%), diabetes mellitus (31.6%), atrial fibrillation (1.6%), and chronic kidney disease (2.2%). These comorbidities are often associated with a high incidence of various complications in HFpEF, and it is not easy to identify a single diagnostic predictor or a cutoff value greater than the risk of these factors. In our study, as in previous studies, a Cox regression analysis identified old age, female sex, high blood pressure, diabetes mellitus, atrial fibrillation, and chronic kidney disease as independent predictors of all-cause mortality (Table 3). The larger the number of these clinical risk factors, the worse the prognosis was (Appendix A). 

### 4.2. Predictive Value of the HFA-PEFF Score in Patients with an HFA-PEFF Score ≥ 2

We observed that the echocardiographic parameters and biomarkers currently used as HFpEF diagnostic tools could also be used to predict the prognosis of AMI HFpEF patients effectively. We compared the C-statistics, IDI, and NRI scores between models with all existing conventional risk factors and models with the addition of high HFA-PEFF scores for predicting the risk of mortality and heart failure associated readmission. Addition of the high HFA-PEFF score to the model significantly improved the predictive accuracy for discrimination (C-index 0.754 [95% CI 0.716–0.791]; *p* = 0.033, IDI 0.007 [95% CI 0–0.014]; *p* = 0.047, respectively), and the probability of reclassification (NRI 0.372 [95% CI 0.227–0.518]; *p* < 0.001) (Table 5). During the long-term follow-up period of 5 years after acute treatment for AMI, the incidence rates of all-cause death, cardiovascular death, and readmission due to heart failure were significantly higher in the high HFA-PEFF score group, while clinical outcomes related to revascularization did not significantly differ between the two groups. Although we used an AMI cohort, the pattern of follow-up clinical event occurrence that was observed was similar to that of previous HFpEF studies because patients underwent successful revascularization and because patients with an LVEF of ≥50% were enrolled. In addition, there was no difference between the two groups in variables, including procedural factors that indicated a complex, high-risk intervention (Table 2). Interestingly, the bleeding event rates (BARC 2, 3, and 5) were significantly higher in the high HFA-PEFF score group even though there was no difference in the use of antiplatelet agents, potent P2Y12 inhibitors, or oral anticoagulants, or the duration of dual antiplatelet agents. This might be associated with a higher prevalence of underlying diseases in the high HFA-PEFF score group. 

### 4.3. Components of the HFA-PEFF Score and Its Meaning

In our cohort, we investigated the prognostic importance of each morphological, functional, and biomarker domain of the HFA-PEFF scoring system. A high biomarker score and a high functional score indicated a significantly higher risk of all-cause mortality in univariate Cox regression analysis. The highest hazard ratio was observed in the biomarker domain compared to the others (HR 2.39, 95% CI 1.57–3.63, *p* < 0.001). NT-proBNP is a well-known biomarker that reflects the prognosis of HFpEF [14,15,16]. LV diastolic dysfunction (e.g., septal E’, lateral E’, and average E/E’), included in the functional domain, plays a central role in the development of HFpEF by impairing relaxation or increasing stiffness [17]. LV diastolic dysfunction can cause an increase in LV filling pressure, promote dyspnea, and increase mortality. [18,19]. In addition, non-diastolic abnormalities such as pulmonary hypertension (PH) and right ventricular (RV) dysfunction may also influence the prognosis of HFpEF. We scored the morphological domains according to the left atrial volume index (LAVI), which had no significant correlation with all-cause mortality. This may be because LAVI may be smaller due to intermittent diastolic pressure overload in early HFpEF, and the structural remodeling that progresses with chronic heart failure was not sufficiently reflected because the echocardiographic parameters were measured within days after the procedure in our study. If the LAVI values obtained several months after the procedure are reflected, the results might be different. In addition, functional indices such as global LA strain or LA conduit strain, which were not assessed in our cohort, might be more appropriate diagnostic parameters.

### 4.4. Appropriateness of the HFA-PEFF Score in the AMI Population

In AMI patients, the NT-proBNP levels are inevitably higher than those in nonischemic HFpEF patients due to ischemic myocardial damage. Therefore, the effect of biomarkers on risk discrimination may be relatively greater than that of functional or morphological factors. Whether the general risk scoring used for general HFpEF diagnosis should be applied equally to the prognosis of AMI patients, particularly the optimal cutoff value and proper test timing of the NT-proBNP level, may be controversial. We used the peak NT-proBNP level tested during hospitalization, and the echocardiographic data acquired a few days after the procedure. According to our results, the current HFA-PEFF score and cutoff value itself showed excellent prognostic evaluation value and reclassification power. 

### 4.5. Treatment of HFpEF

For AMI patients, there is relatively low interest in the HFpEF treatment compared to the HfrEF treatment. This is also due to the absence of clear treatment targets and drugs. There was no convincing evidence-based strategy to improve the prognosis of patients with HfpEF before the PARALLAX clinical trial was announced due to the lack of accurate treatment targets and specified benefit groups [20]. Indeed, the current 2021 ESC guideline recommends the use of diuretics as the only class Ia treatment for HfpEF [5]. In recent studies, attention has been focused on the emergence of empagliflozin and sacubitril/valsartan as new HfpEF drugs. In the case of empagliflozin, the EMPEROR-preserved trial demonstrated a significant benefit for mortality and heart failure readmission rates in patients with symptomatic HF, with an LVEF of >40% and elevated natriuretic peptides [21]. Therefore, the 2022 AHA guideline recommends that SGLT2i can be beneficial for decreasing HF hospitalizations and cardiovascular mortality in patients with HFpEF [22]. In the case of sacubitril/valsartan, although clinical use was authorized by the FDA after further statistical analysis, the primary endpoint of the PARAGON-HF trial, according to the initial design, did not reach statistical significance. Interestingly, in subgroup analysis, for elderly individuals, women, patients with atrial fibrillation, and those with an LVEF of ≤57%, the use of sacubitril/valsartan significantly reduced the rates of mortality and hospitalization due to heart failure [23]. In particular, a significant benefit of sacubitril/valsartan compared with valsartan was observed in patients with an LVEF range of 45–57% (rate ratio 0.78 [95% CI 0.64–0.95]), and in women (rate ratio 0.73 [95% CI 0.59–0.90]) [23,24]. These subgroups were defined by factors known to have a particularly poor prognosis in HFpEF, indicating the importance of distinguishing risk groups during HFpEF. 

### 4.6. Future Perspectives

We believe that the HFA-PEFF score will play a large clinical role in distinguishing high-risk HFpEF patients and between patients responding well to the drug and those who may not. It may be possible to use machine learning to generate indicators that better predict clinical outcomes and responses to treatment by integrating information such as digital imaging data, hemodynamics, conventional risk factors, and new biomarkers [25]. In addition, there have been recent attempts to introduce customized therapy with genetic and clinical phenotyping into HFpEF [26,27]. These challenges of distinguishing HFpEF by various phenotypes or etiologies and genetic causes will increase in the near future, and more appropriate treatments for each classified group may be proposed.

### 4.7. Limitations

First, this study used retrospective observational cohort data. Our findings need validation from prospectively designed research or randomized controlled trials with large populations in the future. Second, the very long period of study could be a confounder due to longitudinal bias despite careful follow-up. However, when the entire registration period was divided by tertile (2004 to 2010, 2011 to 2012, and 2013 to 2014), there was no significant difference between the proportions of High HFA-PEFF score patients for each group. (26.6% vs. 32.7% vs. 30.6%, *p* = 0.211) (Appendix A). In addition, the high HFA-PEFF score group showed a higher cumulative risk for death than the intermediate HFA-PEFF score group consistently in each period (*p* = 0.002, *p* < 0.001, *p* = 0.012). Third, heart failure symptom-related clinical endpoints were not available in this cohort. Fourth, body mass index data that indicate obesity, one of the known risk factors for HFpEF, were excluded from our baseline analysis because of missing values. Fifth, not all parameters of each domain of the HFA-PEFF scoring system were investigated in our study. However, the 2019 ESC Consensus document states that HFA-PEFF scores can be calculated even if all parameters are not acquired with the aim of adding to the practical usefulness of the scores [6].

## 5. Conclusions

Our findings demonstrate that HFA-PEFF scores, which have now been used as an HFpEF diagnostic tool, have significant prognostic and reclassification capabilities beyond conventional clinical risk factors. The high HFA-PEFF score group needs not only conventional heart failure medication but also efforts to actively explore the etiology and better manage related comorbidities.

## Figures and Tables

**Figure 1 jcm-11-04589-f001:**
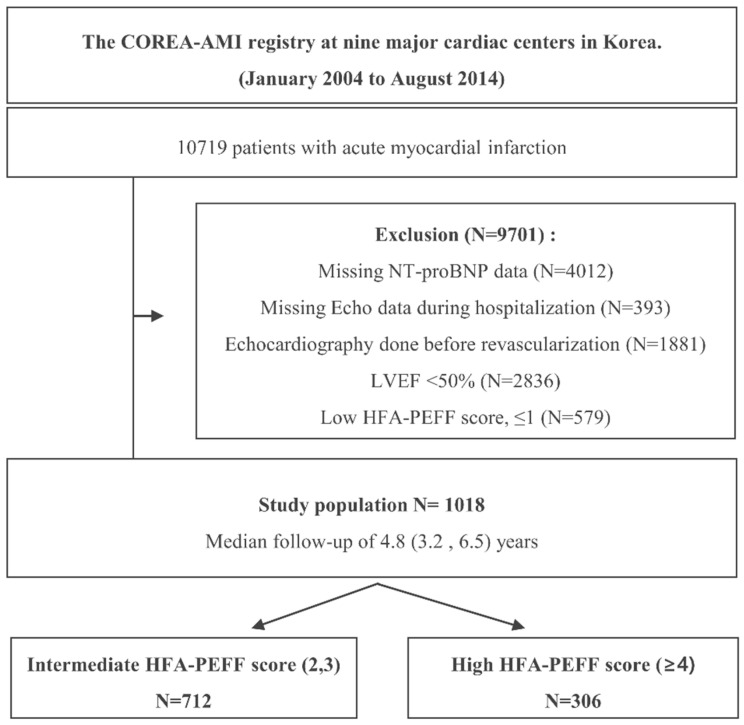
Study flowchart. Abbreviations: NT-proBNP, N-terminal pro b-type natriuretic peptide; LVEF, left ventricle ejection fraction.

**Figure 2 jcm-11-04589-f002:**
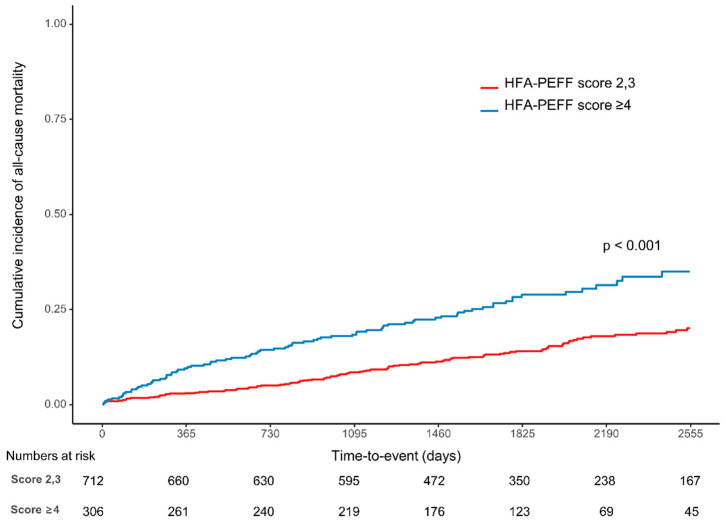
Cumulative incidence of all-cause mortality according to HFA-PEFF scores.

**Figure 3 jcm-11-04589-f003:**
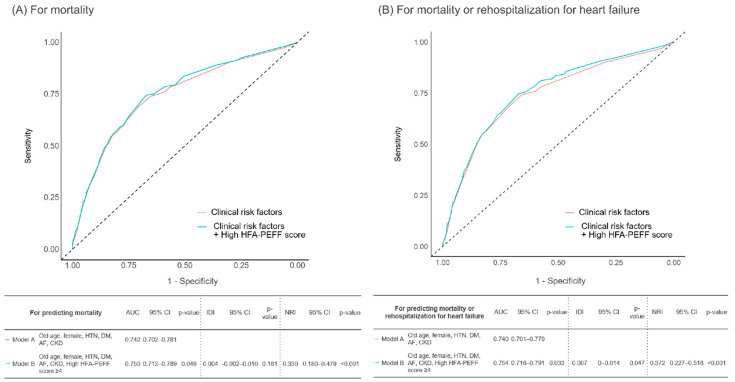
Prognostic impact of HFA-PEFF scores in patients with AMI and an HFA-PEFF score ≥ 2 for predicting (**A**) mortality and (**B**) mortality or rehospitalization for heart failure. Abbreviations: HTN, hypertension; DM, diabetes mellitus; AF, atrial fibrillation; CKD, chronic kidney disease; HFA, heart failure association; AUC, area under the ROC curve; CI, confidence interval; NRI, net reclassification improvement; IDI, integrated discrimination improvement.

**Table 1 jcm-11-04589-t001:** Baseline characteristics.

	Total	Intermediate HFA-PEFF Score (N = 712)	High HFA-PEFF Score (N = 306)	*p*-Value
Clinical characteristics				
Age, yr	65.0 ± 12.3	63.0 ± 12.3	69.7 ± 11.0	<0.001
Female	336 (33.0)	197 (27.7)	139 (45.4)	<0.001
DM	322 (31.6)	209 (29.4)	113 (36.9)	0.021
Hypertension	604 (59.3)	389 (54.6)	215 (70.3)	<0.001
Dyslipidemia	167 (16.4)	117 (16.4)	50 (16.3)	1
History of stroke	72 (7.1)	43 (6.0)	29 (9.5)	0.067
Previous MI	27 (2.7)	19 (2.7)	8 (2.6)	1
Previous PCI	60 (5.9)	36 (5.1)	24 (7.8)	0.113
Previous CABG	5 (0.5)	2 (0.3)	3 (1.0)	0.163
Atrial fibrillation on baseline ECG	16 (1.6)	10 (1.4)	6 (2.0)	0.584
Cancer	52 (5.1)	34 (4.8)	18 (5.9)	0.562
Chronic liver disease	11 (1.1)	6 (0.8)	5 (1.6)	0.321
Chronic lung disease	20 (2.0)	14 (2.0)	6 (2.0)	1
Chronic kidney disease	21 (2.1)	10 (1.4)	11 (3.6)	0.044
KILLIP III or IV	112 (11.0)	70 (9.8)	42 (13.7)	0.087
Cardiogenic shock	37 (3.6)	24 (3.4)	13 (4.2)	0.618
2nd drug-eluting stents	801 (78.7)	552 (77.5)	249 (81.4)	0.197
SBP	130.0 (110.0, 145.0)	130.0 (110.0, 145.0)	130.0 (110.0, 145.3)	0.636
DBP	80.0 (70.0, 90.0)	80.0 (70.0, 90.0)	76.0 (63.8, 86.0)	0.011
HR	76.0 (65.0, 88.0)	76.0 (65.0, 87.0)	78.0 (65.0, 88.0)	0.400
Laboratory findings				
NT-proBNP, ng/mL	499.0 (226.7, 1434.0)	380.0 (172.2, 978.0)	1081.0 (409.0, 2703.0)	<0.001
NT-proBNP in AF, ng/mL	2048.0 (677.2, 5547.5)	1504.3 (414.6, 2950.0)	3219.0 (1249.0, 10734.0)	0.193
NT-proBNP in sinus rhythm, ng/mL	481.0 (225.0, 1424.0)	372.4 (170.8, 970.0)	1063.5 (407.4, 2674.0)	<0.001
Elevated troponin	322 (31.6)	223 (31.3)	99 (32.4)	0.802
CK-MB, peak, ng/mL	73.5 (20.3, 176.7)	78.8 (23.9, 177.6)	58.5 (14.0, 167.4)	0.040
Hemoglobin, mg/dL	13.9 (12.5, 15.2)	14.3 (13.0, 15.4)	13.0 (11.4, 14.3)	<0.001
Platelet, mg/dL	226.0 (186.0, 268.0)	229.5 (189.0, 267.0)	218.5 (184.0, 271.0)	0.201
Creatinine, mg/dL	1.0 (0.8, 1.2)	0.9 (0.8, 1.1)	1.0 (0.8, 1.3)	<0.001
HbA1c, mg/dL	6.1 (5.6, 7.1)	6.0 (5.6, 6.9)	6.3 (5.7, 7.3)	0.078
high-sensitivity CRP, mg/dL	0.5 (0.1, 2.0)	0.4 (0.1, 1.5)	0.8 (0.2, 3.8)	<0.001
Total cholesterol, mg/dL	171.0 (143.0, 201.5)	175.0 (147.0, 206.0)	161.0 (134.3, 191.0)	<0.001
Triglyceride, mg/dL	96.0 (62.0, 138.0)	96.0 (63.0, 143.0)	96.0 (61.0, 133.0)	0.239
High-density lipoprotein, mg/dL	39.0 (33.0, 46.0)	39.7 (34.0, 47.0)	37.0 (31.0, 45.0)	<0.001
Low-density lipoprotein, mg/dL	107.0 (84.0, 133.0)	108.6 (87.0, 136.0)	97.5 (75.5, 128.0)	<0.001
Medication at discharge				
Antiplatelet agent	1009 (99.1)	708 (99.4)	301 (98.4)	0.138
Potent P2Y12 inhibitor	160 (15.7)	118 (16.6)	42 (13.7)	0.149
Beta-blocker	873 (87.4)	610 (87.5)	263 (87.1)	0.322
ACEi or ARB	504 (49.5)	346 (48.6)	158 (51.6)	0.412
Aldosterone antagonist	42 (4.6)	24 (3.7)	18 (6.7)	0.07
Other Diuretics	193 (21.1)	119 (18.4)	74 (27.6)	0.002
Oral anticoagulant	12 (1.2)	5 (0.7)	7 (2.3)	0.051
Statin	929 (95.6)	653 (95.7)	276 (95.2)	0.328
DAPT duration, month	21.3 (13.2)	21.7 (13.1)	20.3 (13.6)	0.112

Data are presented as the n (%) for categorical variables. Continuous variables are presented as the mean ± standard deviation or median (Q1, Q3), according to whether they were normally distributed or not. Elevated troponin is defined as cardiac troponin I ≥ 40 ng/mL or troponin T ≥ 0.1 ng/mL. Other diuretics is defined as use of furosemide, torsemide, or hydrochlorothiazide. The antiplatelet agent includes any of aspirin, clopidogrel, ticagrelor, and prasugrel. The potent P2Y12 inhibitors include ticagrelor or prasugrel. DAPT duration was defined as the number of months in which the patient maintained dual antiplatelet agents during the study period. HFA indicates heart failure association; DM, diabetes mellitus, HTN, hypertension; MI, myocardial infarction; PCI, primary coronary intervention; CABG, coronary artery bypass graft; ECG, electrocardiography; SBP, systolic blood pressure; DBP, diastolic blood pressure; HR, heart rate; NT-proBNP, N-terminal pro b-type natriuretic peptide; BNP, brain natriuretic peptide; CK-MB, creatinine kinase MB isoenzyme; HbA1c, hemoglobin A1C; CRP, C-reactive protein; ACEi; angiotensin-converting enzyme inhibitors; ARB, angiotensin II receptor blockers; DAPT, dual antiplatelet therapy.

**Table 2 jcm-11-04589-t002:** Baseline echocardiographic and angiographic characteristics.

	Total	Intermediate HFA-PEFF Score (N = 712)	High HFA-PEFF Score (N = 306)	*p*-Value
Echocardiographic parameters				
LVEF (%)	58.0 (54.0, 62.8)	58.0 (54.0, 63.3)	57.0 (53.0, 61.0)	<0.001
Left atrial volume index (ml/m^2^)	29.9 (22.5, 39.0)	25.0 (18.7, 31.9)	37.4 (30.5, 45.3)	<0.001
Left ventricular end-systolic diameter (mm)	31.6 (27.2, 35.5)	31.0 (27.0, 35.0)	32.5 (28.3, 36.5)	0.002
Left ventricular end-diastolic diameter (mm)	48.0 (44.0, 52.0)	47.9 (44.0, 51.4)	48.4 (43.6, 53.0)	0.064
Left ventricular end-systolic volume (mL)	32.2 (25.0, 40.3)	32.0 (25.0, 40.0)	33.2 (26.0, 41.1)	0.140
Left ventricular end-diastolic volume (mL)	77.0 (63.3, 94.6)	76.0 (62.4, 94.0)	77.5 (64.4, 96.0)	0.422
E/e’	12.3 (9.8, 16.0)	11.2 (9.1, 13.4)	17.0 (15.0, 20.4)	<0.001
Estimated PASP (mmHg)	28.0 (24.5, 36.0)	27.0 (24.0, 34.0)	31.5 (26.0, 41.0)	<0.001
Angiographic characteristics				
3VD	246 (24.2)	147 (20.6)	99 (32.4)	<0.001
Left main	59 (5.8)	42 (5.9)	17 (5.6)	0.945
Left anterior descending	738 (72.5)	507 (71.2)	231 (75.5)	0.185
Left circumflex	500 (49.1)	325 (45.6)	175 (57.2)	0.001
Right coronary artery	604 (59.3)	403 (56.6)	201 (65.7)	0.008
Total stent number	1.0 (1.0, 2.0)	1.0 (1.0, 2.0)	1.0 (1.0, 2.0)	0.210
Total stent length	28.0 (20.0, 38.0)	28.0 (20.0, 38.0)	28.0 (20.0, 38.8)	0.644
Bifurcation PCI	25 (2.5)	18 (2.5)	7 (2.3)	0.995
Long stenting >60 mm	160 (15.7)	105 (14.7)	55 (18.0)	0.229
CTO	40 (3.9)	33 (4.6)	7 (2.3)	0.111
Restenosis lesion	16 (1.6)	11 (1.5)	5 (1.6)	1
Ostial lesion	36 (3.5)	25 (3.5)	11 (3.6)	1

Data are presented as the n (%) for categorical variables. Continuous variables are presented as the mean ± standard deviation or median (Q1, Q3), according to whether they were normally distributed or not. PASP indicates pulmonary artery systolic pressure; 3VD, three vessel disease; PCI, percutaneous coronary intervention; CTO, chronic total occlusion.

**Table 3 jcm-11-04589-t003:** Risk factors for all-cause death in patients with AMI and an HFA-PEFF score ≥ 2.

				Unadjusted		Multivariable-Adjusted	
	Total (N = 1018)	No Event (N = 817)	Event (N = 201)	HR (95% CI)	*p*-Value	HR (95% CI)	*p*-Value
Echocardiographic scores and parameters							
High HFA-PEFF score group (≥4 points)	306 (30.1)	219 (26.8)	87 (43.3)	2.12 (1.6, 2.8)	<0.001	1.53 (1.15, 2.04)	0.004
High biomarker score: 2	774 (76.0)	598 (73.2)	176 (87.6)	2.39 (1.57, 3.63)	<0.001	1.98 (1.3, 3.03)	0.002
High functional score: 2	327 (32.1)	242 (29.6)	85 (42.3)	1.64 (1.24, 2.18)	<0.001	1.15 (0.86, 1.53)	0.356
High morphological score: 2	130 (37.6)	111 (38.3)	19 (34.5)	0.91 (0.52, 1.59)	0.737	1.01 (0.56, 1.83)	0.963
Conventional clinical risk factors							
Age ≥ 75	245 (24.1)	144 (17.6)	101 (50.2)	4.33 (3.27, 5.73)	<0.001	4.33 (3.23, 5.8)	<0.001
Chronic kidney disease	21 (2.1)	10 (1.2)	11 (5.5)	4.73 (2.57, 8.71)	<0.001	3.97 (2.11, 7.5)	<0.001
Atrial fibrillation	16 (1.6)	8 (1.0)	8 (4.0)	2.98 (1.47, 6.05)	0.003	2.12 (1.04, 4.33)	0.039
Diabetes	322 (31.6)	234 (28.6)	88 (43.8)	1.9 (1.43, 2.51)	<0.001	1.81 (1.35, 2.42)	<0.001
Hypertension	604 (59.3)	458 (56.1)	146 (72.6)	1.97 (1.44, 2.69)	<0.001	1.4 (1.01, 1.92)	0.042
Female	336 (33.0)	258 (31.6)	78 (38.8)	1.35 (1.02, 1.8)	0.038	0.92 (0.69, 1.24)	0.591

Values are the number of events (%) unless otherwise indicated. The variables of multivariate analysis: age ≥75, female, diabetes, hypertension, atrial fibrillation, and chronic kidney disease. HFA indicates heart failure association; HR, hazard ratio; CI, confidence interval.

**Table 4 jcm-11-04589-t004:** Clinical outcomes according to the HFA-PEFF score in patients with AMI suspected of HFpEF.

	Intermediate HFA-PEFF Score (N = 712)	High HFA-PEFF Score (N = 306)	*p*-Value	Unadjusted		Multivariable-Adjusted	
				HR (95% CI)	*p*-Value	HR (95% CI)	*p*-Value
All-cause death	114 (16.0)	87 (28.4)	<0.001	2.12 (1.6, 2.8)	<0.001	1.53 (1.15, 2.04)	0.004
Cardiovascular death	86 (12.1)	66 (21.6)	<0.001	2.15 (1.55, 2.96)	<0.001	1.54 (1.11, 2.15)	0.01
Non-cardiovascular death	28 (3.9)	21 (6.9)	0.065	2.03 (1.15, 3.58)	0.014	1.48 (0.82, 2.65)	0.19
Readmission due to heart failure	11 (1.5)	20 (6.5)	<0.001	4.87 (2.33, 10.21)	<0.001	3.63 (1.69, 7.82)	<0.001
Readmission due to unstable angina	64 (9.0)	23 (7.5)	0.517	0.92 (0.57, 1.48)	0.717	0.77 (0.47, 1.26)	0.299
MI	20 (2.8)	15 (4.9)	0.135	1.04 (0.54, 1.97)	0.914	0.88 (0.45, 1.71)	0.705
Definite or probable ST	39 (5.5)	13 (4.2)	0.508	1.69 (0.55, 5.22)	0.358	1.34 (0.42, 4.24)	0.62
Revascularization	92 (12.9)	38 (12.4)	0.906	1.1 (0.75, 1.61)	0.622	1.06 (0.72, 1.57)	0.759
Ischemic stroke	20 (2.8)	15 (4.9)	0.135	2.02 (1.03, 3.96)	0.041	1.59 (0.8, 3.2)	0.189
BARC 3, or 5 bleeding	53 (7.4)	34 (11.1)	0.072	1.59 (1.03, 2.44)	0.036	1.18 (0.76, 1.85)	0.466

Values are the number of events (%) unless otherwise indicated. The variables of multivariate analysis: age ≥75, female, diabetes, hypertension, atrial fibrillation, chronic kidney disease. HFA indicates heart failure association; HR, hazard ratio; CI, confidence interval; MI, myocardial infarction; ST, stent thrombosis; BARC, Bleeding Academic Research Consortium.

**Table 5 jcm-11-04589-t005:** Effects of variables on the prediction accuracy and risk reclassification of each model (conventional risk factors only vs. conventional clinical risk factors + high HFA-PEFF score model).

Model		C-Index	95% CI	*p*-Value	NRI	95% CI	*p*-Value	IDI	95% CI	*p*-Value
For predicting mortality										
Model A	Old age, female, HTN, DM, AF, CKD	0.742	0.702–0.781							
Model B	Old age, female, HTN, DM, AF, CKD, High HFA-PEFF score (≥4)	0.750	0.712–0.789	0.049	0.330	0.180–0.479	<0.001	0.004	−0.002–0.010	0.161
For predicting mortality and readmission due to heart failure										
Model A	Old age, female, HTN, DM, AF, CKD	0.740	0.701–0.779							
Model B	Old age, female, HTN, DM, AF, CKD, High HFA-PEFF score (≥4)	0.754	0.716–0.791	0.033	0.372	0.227–0.518	<0.001	0.007	0–0.014	0.047

HTN indicates hypertension; DM, diabetes mellitus; AF, atrial fibrillation; CKD, chronic kidney disease; HFA, heart failure association; CI, confidence interval; NRI, net reclassification improvement; IDI, integrated discrimination improvement.

## Data Availability

The datasets used and/or analyzed during the current study are available from the corresponding author on reasonable request.

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
