# Peer review of "Prognostic Impact of the HFA-PEFF Score in Patients with Acute Myocardial Infarction and an Intermediate to High HFA-PEFF Score"

_jcm, 2022, doi:10.3390/jcm11154589_

Round 1
Reviewer 1 Report
The current study was aimed to evaluate the efficacy of HFA-PEFF score in predicting the long-term risks in 1018 patients diagnosed as heart failure with preserved ejection fraction (HFpEF) patients after acute myocardial infarction (AMI). The enrolled subjects were divided according to their HF-PEFF score into intermediate (2-3 points, n=712), and high (4-6 points, n=306) score groups with a median follow-up of 4.8 years. Authors found that a high HFA-PEFF score was an independent predictor of all-cause mortality and they concluded that the HFA-PEFF score is not only a valuable diagnostic tool for HFpEF but can also effectively predict the prognosis in AMI cohort. The study is essential, important and clinically relevant. However, I have some issues that should be addressed.
1. The study was retrospectively performed in nine major cardiac centers in Korea. How did authors guarantee the veracity of individual-based database for the study?
2. There was a big time span of the study about patients’ enrollment, especially for patients with AMI, changes in therapeutic strategies might influence the outcomes for the patients. How did authors manage this issue for their study?
3. In table 1, authors should give the definition of DAPT duration.
4. Was HFA-PEFF Score superior to N-terminal pro b-type natriuretic peptide (NT-proBNP) about in prognostic impact for this cohort study?
Reviewer 2 Report
This study aimed to investigate the efficacy of the HFA-PEFF score in predicting the long-term risks in patients diagnosed with heart failure with preserved ejection fraction (HFpEF) patients after acute myocardial infarction (AMI). The subjects were divided according to their HF-PEFF score into intermediate (2-3 points), and high (4-6 points) score groups. The primary outcome was all-cause mortality. Of 1018 HFpEF patients with AMI, 712 (69.9%), and 306 (30.1%) were classified into the intermediate and high score groups. Over a median follow-up of 4.8 (3.2, 6.5) years, 114(16.0%) and 87(28.4%) patients died in each group. A multivariate Cox regression identified a high HFA-PEFF score as an independent predictor of all-cause mortality [hazard ratio (HR):1.53, 95% CI: 1.15–2.04, p=0.004]. The predictive accuracies for discrimination and reclassification were significantly improved (C-index 0.75 [95%CI 0.712–0.789]; p=0.049, NRI 0.33 [95%CI 0.18–0.479]; p<0.001) upon the addition of a high HFA-PEFF score to clinical risk factors. The model was better at predicting combined events of all-cause mortality and heart failure readmission (C-index 0.754 [95% CI 0.716–0.791]; p=0.033, NRI 0.372 [95%CI 0.227–0.518]; p<0.001). In the AMI cohort, the HFA-PEFF score is not only a valuable diagnostic tool for HFpEF but can also effectively predict the prognosis.
HFA-PEFF is a condition for which effective pharmacological criteria have not yet been standardized, as they are for HFA-REF, so this clinical condition, which remains one of the most important causes of cardiological dyspnoea (NYHA >2), cannot currently be managed pharmacologically outside the now standardized diuretics, the only ones approved for this condition.
The desire to assess prognostic risk in these subjects using a score is of unprecedented importance in the field of dyspnoea of cardiological origin.
The risk in these patients is that there may be overlaps between cardiological pathologies in which echocardiographic evaluation and using bio-markers are undoubtedly important.
The article is well structured and understandable, the diagrams used are very edifying and clarify the prognostic significance of the high score in the proposed score.
I recommend that the following articles be evaluated and, if not already evaluated, proposed as citable:
DOI: 10.1007/s11739-020-02498-7
DOI: 10.1016/j.ejim.2020.04.051
Round 2
Reviewer 1 Report
Thanks for authors' careful response! I have no further comments for the revised manuscript.
Author Response
Thank you for your supportive comments and reply.